# Comparing the Secretomes of Chemorefractory and Chemoresistant Ovarian Cancer Cell Populations

**DOI:** 10.3390/cancers14061418

**Published:** 2022-03-10

**Authors:** Amy H. Lee, Carolina Mejia Peña, Michelle R. Dawson

**Affiliations:** 1Center for Biomedical Engineering, Brown University, Providence, RI 02912, USA; amy_lee1@brown.edu; 2Department of Molecular Biology, Cell Biology and Biochemistry, Brown University, Providence, RI 02912, USA; carolina_mejia_pena@brown.edu

**Keywords:** tumor microenvironment, ovarian cancer, therapeutic and environmental stress, subpopulations and heterogeneity, secretome, extracellular vesicles

## Abstract

**Simple Summary:**

Epithelial ovarian cancer (EOC) is a gynecological disease that is complicated to treat due to its heterogenous nature and because many women develop resistance to various therapeutic strategies. Tumor recurrence can be examined as a two-pronged approach: resistance developed after multiple exposures to frontline anticancer drugs or resistance developed in response to poor microenvironmental conditions, such as hypoxia. Although there are numerous ways to confer chemoresistance, studies have shown that chemoresistant EOC cells release unique secretome profiles that include cytokines, growth factors, and extracellular vesicles (EVs). These secreted factors activate intracellular pathways that contribute to chemoresistance. Secreted EVs transfer biomaterials (including proteins, RNAs, and microRNAs) to other cells, which is critical in cell–cell communication; thus, changes in EV content, in particular exosome miRNAs, have been used to project EOC prognosis. This review examines the feedback loop where chemoresistant EOC cells release unique secretome profiles that confer chemoresistance in normal bystander cells and cancer cells.

**Abstract:**

High-grade serous ovarian cancer (HGSOC) constitutes the majority of all ovarian cancer cases and has staggering rates of both refractory and recurrent disease. While most patients respond to the initial treatment with paclitaxel and platinum-based drugs, up to 25% do not, and of the remaining that do, 75% experience disease recurrence within the subsequent two years. Intrinsic resistance in refractory cases is driven by environmental stressors like tumor hypoxia which alter the tumor microenvironment to promote cancer progression and resistance to anticancer drugs. Recurrent disease describes the acquisition of chemoresistance whereby cancer cells survive the initial exposure to chemotherapy and develop adaptations to enhance their chances of surviving subsequent treatments. Of the environmental stressors cancer cells endure, exposure to hypoxia has been identified as a potent trigger and priming agent for the development of chemoresistance. Both in the presence of the stress of hypoxia or the therapeutic stress of chemotherapy, cancer cells manage to cope and develop adaptations which prime populations to survive in future stress. One adaptation is the modification in the secretome. Chemoresistance is associated with translational reprogramming for increased protein synthesis, ribosome biogenesis, and vesicle trafficking. This leads to increased production of soluble proteins and extracellular vesicles (EVs) involved in autocrine and paracrine signaling processes. Numerous studies have demonstrated that these factors are largely altered between the secretomes of chemosensitive and chemoresistant patients. Such factors include cytokines, growth factors, EVs, and EV-encapsulated microRNAs (miRNAs), which serve to induce invasive molecular, biophysical, and chemoresistant phenotypes in neighboring normal and cancer cells. This review examines the modifications in the secretome of distinct chemoresistant ovarian cancer cell populations and specific secreted factors, which may serve as candidate biomarkers for aggressive and chemoresistant cancers.

## 1. Introduction

Ovarian cancers are broadly categorized based on the tissue of origin into three groups—epithelial, germ cell, and stromal cell [1]. Epithelial ovarian cancer (EOC) is the predominant diagnosis, accounting for 90% of new ovarian cancer cases, and often originates from the fallopian tubes [2]. High-grade serous ovarian cancer (HGSOC) accounts for 75% of all EOC cases, making HGSOC the predominant form of ovarian cancer. HGSOC has a mortality rate of 60% and is often diagnosed at advanced stages, post-metastasis [1]. A major contributing factor to the high mortality rate is that up to 25% of patients experience refractory disease and are inherently resistant to frontline treatment. Furthermore, of the remaining patients that initially respond to treatment, 70% acquire resistance and experience disease recurrence [1,3]. These statistics suggest that refractory and resistant HGSOC are all too common and highlight our gap in understanding how resistance is developed prior to, and as a result of, chemotherapeutic intervention.

Chemoresistance can be classified as either intrinsic or acquired, based on when the resistance is developed. Prior to chemotherapy, refractory cancers develop an intrinsic resistance to anticancer drugs [4] which demonstrates the priming and protective capabilities of the tumor microenvironment (TME). The HGSOC TME is comprised of a host of stromal cells recruited by cancer cells (e.g., fibroblasts, immune cells, endothelial cells) and the extracellular matrix (ECM) developed and maintained by the constant bidirectional communication between the cancer and stromal cell populations. The TME can be protective and protumorigenic; however, as a consequence of a rapidly growing cancer, the TME can also be a hostile environment with limitations in oxygen diffusion, nutrient depletion, and increased mechanical pressure. Each of these stressors can be cytotoxic to cancer cells [5,6,7]; and yet, under the right circumstances, exposure to such stress can serve as a potent trigger to dramatically modify and prime the TME for future insults (Figure 1).

### 1.1. Extrinsic Tme Stressors Promote Intra- and Intercellular Adaptations

For example, in vivo, mechanical stress is felt and exerted by a growing tumor. External compressive stress is experienced by the tumor as the extracellular matrix (ECM), blood and lymphatic vessels resist tumor growth. Internal compressive stress is also developed as cancer cells proliferate within a confined space. Tensile stress is felt more strongly by cells along the periphery of the growing mass. Shear stress is experienced by cells exposed to the flow of interstitial fluid and, in the case of 40% of HGSOC cases, ascites fluid [8,9,10,11]. While our knowledge of the effects of mechanical stress on HGSOC is incomplete, it is established that mechanosensitive cancer cells, that is, cells that can internalize and respond to mechanical stimuli, can thrive or suffer under specific mechanical environments [7,10]. We have shown that HGSOC cells exhibit enhanced proliferation and chemoresistance on softer substrates [7]. Conversely, both 2D and 3D in vitro studies have demonstrated that stiffer substrates and high compressive forces can stunt HGSOC proliferation and promote apoptosis [6,7].

HGSOC cells alter both intracellular mechanisms and the surrounding stromal populations in response to mechanical stress. Indeed, HGSOC cells that are exposed to shear and compression stress in vitro develop stem cell-like properties with increased metastatic potential and chemoresistance [10]. HGSOC cells also have the ability to activate fibroblasts into cancer-associated fibroblasts (CAFs) either through enhanced paracrine signaling or direct physical interactions [12,13]. CAFs then play a crucial role in maintaining the TME by increasing the secretion of ECM components such as collagen and fibronectin or ECM-remodeling proteins such as crosslinking proteins (lysyl oxidases) or matrix-degrading factors (matrix metalloproteinases (MMPs). CAFs, therefore, have the ability to change both the chemical and mechanical composition of the TME to enhance tumor growth and cancer cell invasion [14,15,16].

Similarly to mechanical stress, stress from nutrient depletion in the TME also has the ability to induce potent changes to cancer and stromal populations that have a lasting impact on disease progression. In the presence of extrinsic stressors such as the depletion of nutrients including glucose and amino acids, DNA damage, hypoxia, and reactive oxygen species (ROS), cellular homeostasis is compromised and thus triggers a series of responses that can ultimately drive cells to reorganize their bioenergetic process in a process called metabolic reprogramming. Examples of metabolic reprogramming include but are not limited to increased mitochondrial production, induction of the unfolded protein response (UPR), integrated stress response, and heat shock response [17,18].

The kind and degree of metabolic reprogramming which a cancer cell undergoes is tuned to the present stress. Heterogeneity in metabolic reprogramming within a single tumor is, therefore, highly likely given the range of local environments developed as a result of growth in 3D space. As cancer cells proliferate, recruit stromal cells, and promote matrix remodeling and angiogenesis, diffusion limitations are generated and gradients of nutrients and oxygen availability evolve as the volume and density of the tumor changes over time [6,19]. The metabolic landscape within a single tumor is, therefore, complex. While our understanding of metabolic reprogramming in ovarian cancer is incomplete, metabolic signatures that span cancer types have been identified and point to canonical pathways that are often aberrantly regulated.

### 1.2. Metabolic Reprogramming as a Priming Mechanism in Response to TME-Associated Stress

One such pathway is glycolysis and specifically the uptake of glucose, increase in glycolysis, and preferential fermentation to lactate despite the presence of oxygen, which is also known as the Warburg effect [20]. An increase in anaerobic glycolysis is critical under hypoxic conditions to maintain necessary ATP levels [21]. Alternatively, certain tumor subtypes exhibit a preference for oxidative phosphorylation (OXPHOS) [22]. In a study characterizing the metabolism of 127 clinical HGSOC samples and 14 ovarian cancer cell lines, Gentric et al. demonstrated that there was inherent metabolic heterogeneity among both samples and that they could be clustered into a high-OXPHOS and low-OXPHOS groups. High-OXPHOS samples relied on the tricarboxylic acid (TCA) cycle while low-OXPHOS cells mainly relied on glycolysis. The samples and cell lines categorized as high-OXPHOS had increased electron transport chain (ETC) synthesis, ATP production, basal respiration rate, and mitochondrial content. Importantly, these populations, labeled as high-OXPHOS, also exhibited enhanced chemosensitivity to taxanes and platinum-based drugs [22].

Cancer cells under nutrient deprivation and other environmental stressors are also known to have a particular reliance on glutamine and glutaminolysis. Indeed, glutamine not only serves as an alternative carbon source to fuel the TCA cycle and OXPHOS, but is also a key determinant of the ability of a cell to maintain redox homeostasis via the production of glutathione [23,24,25,26]. Critically, cancer cells also rely on glutamine for its role inducing and coordinating metabolic reprogramming across bioenergetic processes under the stress of nutrient deprivation or chemotherapy [27,28].

Upregulation of fatty acid (FA) uptake and metabolism have emerged as critical elements to metabolic reprogramming in HGSOC. Both the primary tumor site and metastatic niche of the omentum are abundant in FAs as a result of malignant peritoneal fluid buildup (ascites) and the secretome of adipocytes [29]. FAs can serve as alternative carbon sources to fuel the TCA cycle and OXPHOS for ATP when other carbon sources are not available [30,31,32,33]. FAs also serve as key secondary signaling molecules to enhance the proliferation and survival of cancer cells, a well-studied example being phosphatidylinositols and specifically PIP_3_ and its role in the activation of the AKT–PI3K pathway [34]. Importantly, inhibition of FA uptake and metabolism greatly reduce HGSOC resistance to anoikis and overall progression [35,36].

Aberrant regulation of, and reliance on multiple bioenergetic and biosynthetic pathways provide cancer cells with alternative carbon sources to generate ATP and other critical molecules. Coordination of the induction and maintenance of these pathways is complex and essential to achieving cellular homeostasis. Autophagy is responsible for the degradation of compromised or excess proteins and organelles [37]. Briefly, autophagy proteins (ATGs) come together to form a double-membraned vesicle (autophagosome) labeled with the membrane-associated protein, LC3-II. LC3-II is then recognized by adaptor proteins which traffic specific substrates for degradation. Lysosomes then fuse with autophagosomes, a process regulated by SNARE proteins and small GTPases, and the engulfed substrates are broken down by lysosomal enzymes [38]. In the context of cancer under environmental stress, autophagy can generate pools of metabolites (for example, amino acids from proteins, FAs from lipids, and sugars from DNA) that feed into biosynthesis pathways (TCA cycle and glycolysis). By modulating available metabolite pools, autophagy regulates the impact any one pathway has on metabolic stability, thus conferring metabolic plasticity and overall resilience [37].

The selective autophagy of mitochondria, or mitophagy, provides another means of metabolic reprogramming. Mitochondria are tagged for degradation mainly when OXPHOS is compromised, which can be a consequence of the accumulation of ROS [39]. Oxidative stress is accrued when ROS levels surpass the capacity of antioxidant systems to remove ROS—a state often associated with hypoxia [40,41]. Mitophagy serves as such a mechanism to attenuate oxidative stress and reestablish redox homeostasis by reducing the production of ROS and the extent of oxidative damage [40,42]. In addition to mitophagy, mitochondrial fission and fusion can be modulated to reorganize and adapt mitochondrial networks in response to hypoxia. Indeed, HGSOC cells exposed to peritoneal-like hypoxic conditions had elevated levels of ROS and exhibited enhanced fragmentation [41]. Importantly, studies have shown that cancer cells primed with such mitochondrial stress have enhanced chemosensitivity [43,44]. Priming by oxidative stress has also been demonstrated to increase the apoptotic threshold by overexpressing prosurvival factors and thus conferring chemoresistance [45,46]. 

Environmental stressors promote metabolic reprogramming as an intracellular adaptation to cope with such stress. Extrinsic stress also enhances paracrine signaling which ultimately impacts the malignancy and resiliency of the TME [47]. The HGSOC secretome is an abundant source of cytokines, growth factors, and extracellular vesicles that guide critical proteins, transcription factors, and miRNAs through complex extracellular environments [47,48]. Buildup of such a malignant secretome is, unfortunately, common in late-stage HGSOC and referred to as ascites [47]. Although these proteins, nucleic acids, lipids, and vesicles are constantly being secreted, their release can be triggered and altered by multiple stimuli [11]. These can include TME stressors, such as hypoxia, interstitial pressure, and therapy-induced damage [49,50]. These stimuli alter intracellular pathways in ways that are reflected in the molecular content of the cytosol and biomaterials (proteins, RNAs, and miRNAs) that are packaged into multivesicular bodies and their secreted exosomes [51,52]. Therefore, differentially regulated miRNAs can serve as biomarkers for aggressive and even drug-resistant HGSOCs prior to the administration of chemotherapy.

### 1.3. Chemorefractory HGSOC Highlights the Priming Capabilities of the TME

The recommended chemotherapy treatment for advanced-stage HGSOC after diagnosis is a course of frontline drugs carboplatin and paclitaxel [53]. Platinum-based drugs form DNA adducts through crosslinking which prevents DNA synthesis and leads to the accumulation of double-strand breaks (DSBs). The inability to synthesize DNA or repair the resulting DNA damage can cause apoptosis [54]. Antimitotic agent paclitaxel functions by binding to *β*-tubulin subunits, thus stabilizing microtubules (MTs). Unable to depolymerize or reorganize MTs, mitosis is stalled as cells attempt to undergo cytokinesis, and apoptosis is subsequently activated [55]. Several modes of resistance have been identified in in vitro studies to either paclitaxel or platinum-based drugs, and studies using clinical samples have been critical in identifying the common traits of cancer cells; however, treatment efficacy and future drug development would greatly benefit from the ability to differentiate between resistant populations that are primed by the TME and those that are not. Critically, by characterizing 22 matched pre- and post-neoadjuvant chemotherapy-treated HGSOC patient samples, Zhang et al. demonstrated that treatment enriches subpopulations with an initially increased transcriptomic stress response and primes HGSOC cells to resist chemotherapy [56]. Kan et al. similarly identified that relapse-initiating HGSOC cells can largely originate from a subpopulation early in tumorigenesis with a high-stress signature. Relapse-initiating cells also developed over the course of HGSOC progression [57]. By identifying the intrinsically resistant populations in refractory HGSOC, subsequent chemotherapeutic interventions can be designed and optimized to target known vulnerabilities of these distinct populations. 

In this review, we discuss how the prominent TME stressor of hypoxia and therapy-induced stress drive HGSOC cell populations to secrete a unique secretome content that both fosters chemoresistant populations and promotes invasive phenotypes in neighboring populations.

## 2. Differences in the Development of Chemorefractory and Chemoresistant HGSOC Populations

### 2.1. Hypoxia Confers Resistance in Refractory HGSOC

While 25% of HGSOC cases are refractory and exhibit intrinsic resistance to chemotherapies, studies with a specific focus on refractory HGSOC are limited, with most using treatment-naïve samples [58]. By characterizing pre- and post-chemotherapy HGSOC patient samples using single-cell RNAseq, whole genome sequencing, and DNA copy number analysis, groups have found either no recurrent genomic changes in subpopulations primed to endure chemotherapy, *BRCA1/2* mutations or amplification of 19q12, containing cyclin E (CCNE1) (Table 2) [56,58,59]. 

Indeed, these multipronged studies emphasize that intrinsically resistant subpopulations are most unique in their transcriptional profiles. Furthermore, such profiles consistently describe a stress-associated state which primes cancer cells to endure exposure to chemotherapy. These results highlight the potent effects of environmental stressors and the role of subsequent coping mechanisms in generating a resilient HGSOC population.

One of the most potent stressors in the TME is hypoxia. As cancer cells proliferate and recruit neighboring cells, the TME evolves in 3D space and spatial heterogeneity is developed. As a result, oxygen gradients are formed by diffusion limitations throughout the 3D growth [6,19], thereby creating a range of local environments. It follows, then, that the responses cells employ to cope with stress are nonuniform and produce heterogeneity in proliferative capability, migratory probability, metabolism, and chemoresistance.

In normoxic conditions, the hypoxia-inducible factor alpha (HIF-*α*) subunit is hydroxylated and recognized by the von Hippel–Lindau tumor suppressor protein (pVHL) for degradation via the ubiquitin–proteasome pathway. With lower concentrations of oxygen, the degradation of HIF-*α* is stunted; it then dimerizes with the constitutive HIF-1*β*. The HIF complex can then act upon a wide range of target genes [60,61]. While we continue reviewing how chemoresistance is developed, in part, as a result of HIF-1*α* activity, we acknowledge that HIF-1*α* activity is not exclusively triggered by low oxygen concentrations. Several studies have revealed oxygen-independent pathways of HIF-1*α* activation such as the accumulation of metabolites including lactate, pyruvate, and succinate [62,63,64,65,66].

Clinically, overexpression of the HIF-1*α* isomer regardless of the p53 status highly correlates with poor prognosis and resistance to platinum-based drugs in ovarian cancer [67]. Furthermore, exposure to hypoxia either prior to or in conjunction with treatment increases resistance to both frontline chemotherapeutics cisplatin and paclitaxel [68,69]. While several mechanisms are employed by cancer cells to cope with hypoxic conditions, a consistent overarching response is to adopt a flexible stem-like phenotype—a process largely regulated by the HIF pathway [70,71]. The epithelial-to-mesenchymal transition (EMT) broadly describes a series of molecular events whereby differentiated epithelial cells elongate and lose their cellular adhesions and develop a less differentiated and more motile mesenchymal phenotype with increased extracellular matrix adhesions [71]. As such, EMT is implicated in stem cell interactions, embryogenesis, wound healing, and cancer metastasis [72,73]. In the context of ovarian cancer, EMT is critical for the ability of cancer cells to disassociate from the primary site, resist anoikis, and migrate to the secondary site. Once cancer cells reach the secondary tumor site, they undergo a mesenchymal-to-epithelial transition (MET) to establish cellular adhesions important in the growth of metastatic tumors [74,75]. 

Given the range of oxygen availability throughout the TME, cancer cells that are exposed to lower oxygen concentrations are driven towards EMT. However, depending on the tissue and the context of extracellular signals, some epithelial cancer cells lose only some epithelial characteristics, demonstrating both mesenchymal and epithelial characteristics [76]. This phenomenon is denoted as partial EMT (pEMT) and is frequently identified across many cancer types and depends on the frequency and degree of hypoxia exposure [68,77,78].

HIF-regulated (p)EMT serves as a mechanism to induce and sustain a flexible resilient cell state. Indeed, hypoxia has been shown to prime cancer cells with a ROS-resistant phenotype which is sustained after the initial exposure and contributes to survival during metastasis [79]. Furthermore, HIF-1*α* induces autophagy and is critical for the maintenance of cancer stem cell (CSC) populations in ovarian cancer [80,81]. HIF-1*α* can also directly promote CSC traits by activating the nuclear factor (NF)-κB transcription factor, which is critical in inflammation [82]. In addition to HIF-1*α*, other signaling axes such as NOTCH1–SOX2 and the UPR have also been identified as the key modulators of the induction of autophagy and CSC maintenance under hypoxic conditions [83,84]. One of the advantages of adopting a stem cell-like phenotype is developing plastic metabolism through the induction of autophagy and cancer stem cell-like phenotypes. While the precise cascade of alterations to bioenergetic pathways in ovarian CSCs is still being unraveled, several studies have demonstrated that ovarian CSCs have increased glycolysis, OXPHOS, and altered lipid metabolism which can then, in turn, help them resist nutrient deprivation to survive adverse conditions in the tumor microenvironment [85,86,87]. Lastly, ovarian CSCs that are primed by surviving hypoxic conditions are more aggressive when returned to normoxic conditions as evidenced by enhanced proliferation, migration, and colony formation [88].

Studies have demonstrated that hypoxic conditions can also promote proliferation and inhibit apoptosis in ovarian cancer [89]. STAT3 is implicated in several proliferation and apoptosis signaling axes and is often aberrantly activated in ovarian cancer [90]. STAT3 activity is induced by hypoxia and not only regulates proliferation under hypoxic conditions, but also confers both cisplatin and paclitaxel resistance, in part by priming ovarian cancer stem cells to survive in adverse conditions [49,91,92]. Furthermore, silencing STAT3 via siRNA or a chemical inhibitor reverses hypoxia-dependent resistance [93,94].

Overall, hypoxia elicits a metabolically flexible stem cell-like state by the induction of EMT. The subsequent induction of autophagy and metabolic reprogramming are not only critical to the maintenance of such a cell state, but also confer resistance to the present environmental stress and future stress of chemotherapy. Herein, we discuss the associated changes in the secretome of HGSOC cells primed by the stress of their TME.

### 2.2. Hypoxia Alters HGSOC Secretome Profile

The tumor secretome consists of a wide range of proteins, growth factors, and metabolites that play important roles in cell–cell and cell–matrix interactions [95]. Aberrant autocrine and paracrine factors found in the tumor secretome can serve as prognostic and diagnostic HGSOC markers [96]. Hallmark TME stresses (i.e., highly dysfunctional vasculature with leaky, compressed blood vessels and poor microcirculation, which results in reduced oxygen concentrations) can result in heterogenous secretome profiles [97,98]. Secretome changes are of particular importance in the context of HGSOC because patients present with large accumulations of ascites fluid (potent source of cell-secreted factors or the secretome) that are unique to these cancers and metabolic conditions in the peritoneal cavity.

Clinical studies have found that ascites components that are largely altered in patients with chemosensitive tumors vs. patients with drug-resistant tumors include (i) cytokines, (ii) growth factors, and (iii) extracellular vesicles (EVs) [99,100,101,102,103]. EVs—i.e., exosomes, microvesicles, and oncosomes—are highly involved and play a critical role in transferring drug-resistance phenotypes during cell–cell communication. Ascites fluids harvested from HGSOC patients are rich with circulating vesicles and show atypical levels of secreted cytokines and growth factors that are linked to protumorigenic pathways. Numerous cytokines and growth factors, such as interleukin 6 (IL-6), interleukin 8 (IL-8), transforming growth factor-beta (TGF-β), epidermal growth factor (EGF), and vascular endothelial growth factor (VEGF), are upregulated and can alter the TME to promote chemoresistance by activating antiapoptotic and prosurvival signaling pathways [104,105,106]. Clinical studies have revealed that patient ascites contain ~40–500 times more proinflammatory cytokines compared to serum [107]. EVs have more recently emerged as a novel mechanism of enhancing drug resistance via cell–cell communication [108]. These circulating vesicles encapsulate and transfer cellular cargo, along with cytokines, growth factors, and other non-protein content, that can alter recipient cells to develop potent drug-resistant phenotypes [109].

#### 2.2.1. Cytokines

Cytokines—i.e., IL-6, IL-8, IL-11, IL-27, IL-31—are small immunological proteins important in autocrine, paracrine, and endocrine signaling processes that influence inflammation, cell growth and proliferation, cell and matrix interactions, and disease progression in cancer [110]. IL-11, IL-27, and IL-31 are members of the IL-6 family. IL-11 has been reported as an important anti-inflammatory or tumor-promoting cytokine. IL-11 has also been particularly linked to poor prognosis in cancers that possess epithelial traits [99]. However, the specific role of IL-11 in HGSOC remains unclear as IL-11 expression is low in this specific cancer [111]. Similarly to IL-11, IL-27 takes on a dual role in the context of the TME. However, it has been highlighted that IL-27 suppresses SKOV3 cell proliferation by simultaneously activating STAT3 and inhibiting the Akt pathway [112]. Elevated IL-31 levels have been correlated to poor prognosis [113]. Further, IL-31 has been shown to enhance mesenchymal HGSOC cell phenotypes such as proliferation, migration, invasion, and survival. IL-17 is a proinflammatory cytokine that is largely produced by T helper cells and macrophages. Studies have shown that IL-17 assists the renewal of cancer stem-like cells, thereby driving HGSOC tumorigenesis [114].

Cytokines are secreted by immune cells and other cells in the tumor, including HGSOC cells and surrounding primary TME cells (i.e., stromal, endothelial, epithelial, mesothelial) [11,100]. Cytokines mediate intercellular communication between immune and nonimmune cells in the TME. This communication can lead to T cell activation and macrophage differentiation [115]. It has been further established that individual inflammatory cytokines or their cohorts work collectively in activating potent cell phenotypes that assist in conferring chemotherapy resistance and promoting ovarian cancer immune evasion. Tumor cells release these immunosuppressive cytokines, which can activate tumor-associated macrophages and reprogram cells that used to be immunostimulatory to immunosuppressive [116]. These cells then, in turn, secrete immunoinhibitory cytokines that support tumor cell survival, metastasis, and cytotoxic T cell recruitment [117]. In fact, high-grade serous HGSOC are often regarded as immunologically “cold” tumors.

These cytokines serve as important bio- or predictive markers. Numerous in vitro studies examining cell types, such as IGROV-1, PEO1, SKOV3, and OVCAR3, have reported that tumor necrosis factor-alpha (TNF-α), IL-6, and IL-8 act as potent paracrine and autocrine signaling molecules that regulate cancer cell invasion, proliferation, and bulk tumor growth. These phenotypic differences contribute directly to promoting intrinsic chemoresistance [118,119].

Tumor hypoxia can activate transcription factors that are responsible for elevating intracellular cytokine expression and release profiles. Elevated cytokine release from HGSOC TME cells amplifies the autocrine and paracrine feedback loop, which is involved in immune cell recruitment. Cytokines are potent mediators of immune cell homing to tumors and metastatic tissues. This subsequently highlights that hypoxia-induced cytokines serve as proinflammatory factors that are critical in establishing chemorefractory phenotypes. For example, in vitro studies have shown that hypoxia directly modulates activator protein-1 (AP-1) and NF-ĸβ in HGSOC cells. AP-1 and NF-ĸβ conjunctively play important roles in secreting higher IL-8 levels in SKOV and Hey8 cell lines [120]. Hypoxic conditions, low pH, and poor vascularization in the TME further contribute to the abnormal cytokine profiles. HGSOC cells cultured in in vitro acidic conditions (pH 6.6) demonstrated increased IL-8 secretion, which was associated with the activation of AP-1 and NF-ĸβ and the development of chemoresistance [121].

TNF-α also plays a key role in regulating local and distal invasion, angiogenesis, and metastasis in numerous cancers, particularly in HGSOC. Similarly to IL-6 and IL-8 profiles, TNF-α is found at high circulating concentrations under hypoxic conditions and in HGSOC patient TMEs. TNF-α plays pivotal roles in autocrine and paracrine signaling mechanisms that activate pathways important in invasive and chemoresistant cancer cell behaviors. For example, TNF-α activates NF-ĸβ and further canonical pathways [122]. Activation of NF-ĸβ subsequently reduces the p53 transcription factor activity; p53 is a tumor suppressor that halts cell cycle progression and cell division in response to DNA damage or genetic instability [123]. Mutations in p53 and loss of functional p53 activity trigger invasive and chemoresistant cell behavior [124].

Proinflammatory cytokines secreted in response to hypoxic conditions are important paracrine and autocrine factors that regulate signaling pathways responsible for cancer cell proliferation, intrinsic chemoresistance, and invasion in local and distal tumor sites.

#### 2.2.2. Growth Factors

The HGSOC secretome is enriched with growth factors that support and maintain cells that have detached from the primary tumor and attached at the secondary site. The repertoire of growth factors and growth factor receptors is altered between HGSOC patient and normal donor tissues and blood and can aid in conferring chemoresistance [124,125]. Anticancer drugs and therapies not only influence the growth factor profiles in the secretome, but potent hypoxic conditions also contribute to this abnormal profile of circulating growth factors [126]. As a homeostatic response to hypoxic TME conditions, HGSOC cells often secrete increased concentrations of EGF. EGF further maintains tumor hypoxia by elevating HIF-1α expression [127]. Simultaneous in vitro elevation of EGF and HIF-1α promote EMT phenotypes in SKOV3 and OVCAR5 cells, which are associated with reduced E-cadherin and increased Snail and Slug expression [127]. Further, HIF-1α increases VEGF expression [128]. VEGF is essential in angiogenesis and ultimately promotes tumorigenesis through cell migration, proliferation, and survival [129]. Overall, hypoxic conditions activate transcription factors that trigger aberrant growth factor secretion.

#### 2.2.3. EVs

TME stresses, particularly hypoxia, increase EV biogenesis from cancer cells, while also regulating EV protein and miRNA synthesis; yet, it is not well-understood how hypoxia directly accelerates EV secretion. A substantial body of in vitro evidence has linked HIF-1α activation to elevated proteins responsible for EV secretion, such as Rab proteins that regulate intracellular vesicle transport [130,131].

A more recent in vitro study has further demonstrated that hypoxia triggers CAOV3 cells to release small EVs that confer carboplatin resistance to neighboring cancer cells. Specifically, EVs released under these hypoxic conditions alter glycolytic and fatty acid synthesis pathways that support chemoresistance in recipient, normoxic HGSOC cells [103]. More importantly, a cohort of these identified EV-associated glycolytic proteins were patient-specific and can be further used as predictive markers for refractory HGSOC [103].

### 2.3. Hypoxia Alters HGSOC Exosomes

Exosomes are a subcategory of EVs (40–160 nm diameter) that play crucial roles in intercellular communication and chemoresistance. Exosomes play pivotal roles in cell–cell interactions, often altering physical phenotypes and signaling pathways in recipient cells. Their cholesterol-rich membranes allow exosomes to serve as stable vehicles during intercellular communication and prevent enzymatic degradation of sensitive protein and RNA content. These exosomes are packaged with functional biomolecules, such as proteins, RNAs, and miRNAs, during intraluminal vesicle transport; the exosome content is highly representative of parent cells (i.e., drug-resistant HGSOC cells) and HGSOC tumor microenvironment [132]. This allows exosomes and encapsulated cargo to act as useful biomarkers for malignant and invasive HGSOCs [132]. In vitro evidence with human cervix carcinoma cells has highlighted that chemoresistance alters proteins that are essential in the endosomal pathway, subsequently regulating exosome biogenesis [133].

Additionally, hallmark primary TME conditions that accompany chemorefractory development, such as hypoxia and low pH, alter and accelerate exosome release mechanisms. In vitro cancer cells (i.e., various metastatic grades of HGSOC, breast, pancreatic, and lung cancer cells) and cancer-associated stromal cells (i.e., fibroblasts, stem cells, myeloid cells) cultured under hypoxic conditions secreted more exosomes than cells cultured under normoxic conditions. This rapid exosome secretion may be attributed to the upregulation or activation of transcription factor genes under hypoxic stress [49,134,135,136]. These transcription factors, including tetraspanins, soluble *N*-ethylmaleimide-sensitive factor attachment protein receptors (SNAREs), tethering proteins, and Rabs, regulated exosome biogenesis and secretion proteins. HIF-1α activates Rab22a, an essential protein during exosome secretion. Rab22a colocalizes to budding vesicular membranes [137]. Actins, along with other major cytoskeletal proteins, are critical in transporting extracellular vesicles from the cytoplasm to the plasma membrane. Hypoxia can also alter actin filament organization. In vitro studies with A2780 cells have shown that hypoxia positively induces ROCK, a regulator of actin dynamics, to increase exosome biogenesis and secretion [138]. These secreted exosomes encapsulated aberrant levels of HIF-1α, interleukins, and matrix degradation enzymes. Exosomes subsequently propagated EMT phenotypes and conferred chemorefractory resistance in recipient cells [139]. 

### 2.4. Acquired Chemoresistance Is Achieved through Drug-Specific Adaptations and Induction of Therapy-Induced Senescence

Cancer cells that manage to cope with hostile TME conditions are more likely to survive future chemotherapeutic treatments. The TME can increase the apoptotic threshold in cancer cells as well as induce a dormant stem-like phenotype. Additionally, the TME promotes modifications in the HGSOC secretome which fosters a protective phenotype in neighboring cells and reinforces resistance. Evasion of subsequent chemotherapy treatment can promote additional drug-specific modes of resistance.

Ovarian cancer cells that survive paclitaxel treatment often develop altered cytoskeletal dynamics [55]. Cells that survive paclitaxel have been shown to have molecular alterations in tubulin variants and post-translational tubulin modifications that hinder paclitaxel from properly binding MTs [140]. Dynamics of MT growth and catastrophe are also altered in response to paclitaxel treatment and have been shown to enable ovarian cancer cells to more quickly attach and migrate on polyacrylamide substrates through increased traction stress generation [7]. In addition to cytoskeletal changes, HGSOC cells can employ efflux pumps to cope with paclitaxel treatment. Paclitaxel is a substrate of efflux pump P-glycoprotein (P-gp) [141,142]. P-gp is encoded by gene MDR1, and across several studies, MDR1 expression or P-gp structure has been enhanced or modified, respectively, in ovarian cancer cells after chemotherapy [1,59].

### 2.5. Therapy-Induced Senescence and Escape as a Mechanism for Recurrent Disease

Paclitaxel and platinum-based drugs can also elicit similar changes seen in response to TME stressors [140,143,144,145,146,147]. Post-chemotherapeutic intervention, these collective changes, including the induction of autophagy, metabolic reprogramming, EMT, and enhanced paracrine signaling, are indicators of therapy-induced senescence (TIS). In addition to the aforementioned changes, cancer cells with a TIS profile have characteristics associated with cellular senescence such as cell cycle arrest and compromised nuclear integrity. TIS has emerged as a key contributor to dormant tumors and recurrent disease [148]. We should note that while dormancy has been associated with a quiescent cell profile, senescence more accurately describes the overall trajectory and molecular changes sustained by cells that escape dormancy and fuel recurrent disease. A quiescent cell will revert to a prestress phenotype and resume proliferation once the stressor (intrinsic or extrinsic) has been removed; however, a senescent cell does not necessarily revert or begin to proliferate under favorable conditions and, importantly, retains some of the adaptations developed in response to the stress [148,149].

Cancer cells can also undergo senescence in the absence of therapy, in a process named oncogene-induced senescence (OIS). OIS is driven by oncogenic stress (for example, DNA damage) that is generated by the aberrant expression or regulation of oncogenes or tumor suppressors (including, but not limited to, Ras, Raf, Akt, PTEN) [150]. The prerequisite to both OIS and TIS is activation of the DNA damage repair (DDR) response [151]. While in OIS the DDR response is activated to cope with DNA damage caused by telomere attrition and loss of tumor suppressors, the DNA damage leading to TIS can be caused directly by DNA crosslinking properties of platinum-based drugs (cisplatin or carboplatin) as well as the associated DNA damage caused by prolonged paclitaxel-induced mitotic arrest [151,152,153,154].

Defective DDR is a hallmark of late-stage ovarian cancers. Platinum-based drugs elicit DNA damage and require cancer cells to either enhance or modify the existing DNA damage repair (DDR) mechanisms [155]. For example, ovarian cancer cells that survive carboplatin treatment often have defective nonhomologous end joining (NHEJ) repair mechanisms [156]. Tumor suppressor genes *BRAC1/2* which are critical in the DDR also have consistent mutations in therapy-resistant patients [59]. Upstream of DDR mechanisms, epigenetic modifications such as DNA methylation and platination are also implicated in cisplatin resistance [155,157].

Cells that are exposed to high concentrations of anticancer drugs may undergo mitotic slippage or endoreplication to form polyploid giant cancer cells (PGCCs), which are multinucleated cancer cells that have high DNA content and multidrug resistance [153,158]. The PGCC phenotype is associated TIS; thus, PGCCs remain metabolically active but no longer undergo mitosis. PGCCs are capable of undergoing amitotic division through depolyploidization, which leads to the formation of diploid cancer cells that are capable of metastasis. These PGCC daughter cells also acquire the PGCC drug-resistant phenotype [153].

Whether mechanisms promoted by the TME, such as EMT, and TIS are mutually exclusive or molecularly linked is unclear; however, recent evidence demonstrates that they have several shared regulators and confer chemoresistance to cancer cells [159]. The implications of TIS and senescence more broadly in cancer progression are complex as there is evidence that suggests both pro- and antitumorigenic effects [151]. While senescence can stunt the proliferation of cancer cells, such dormancy can be protective by decreasing the efficacy of antimitotic drugs. Conversely, escape from dormancy, a process that relies on EMT, greatly contributes to disease recurrence [160]. TIS-associated induction of autophagy and stemness have been shown to drive disease recurrence [148,161]. Importantly, TIS promotes potent changes in the secretome of cancer and stromal populations known as the senescence-associated secretory phenotype (SASP) [162]. Herein, we discuss how the HGSOC SASP bolsters chemoresistance in cancer cells and fosters a protective TME.

### 2.6. Therapy-Induced Chemoresistance Alters the HGSOC Secretome

HGSOC cells exposed to chemotherapeutics release unique secretome components that can confer chemoresistance to surrounding cells and can assist in developing a more invasive TME. Cells that survive therapeutic taxane or platinum-based drugs through TIS can further activate cells to secrete a more potent senescence-associated secretome that aids in chemoresistance development [50]. Similarly to other senescent cells, tumor cells that exhibit TIS remain metabolically active and develop altered secretome profiles. This unique secretome promotes chemoresistance to neighboring tumor-associated cells through the production of cytokines, growth factors, and EVs that promote cell survival.

#### 2.6.1. Cytokines

Women diagnosed with HGSOC have consistently shown elevated circulating levels of IL-6, IL-8, and TNF-α. Clinical data have indicated that IL-6 and IL-8 are significantly amplified between benign ovarian tumors and HGSOCs [163,164]. Studies have shown that innate and activated immune cells are largely responsible for cytokine secretion. Immune cells that contribute to cytokine production include macrophages, T cells (i.e., CD8+ and CD4+), lymphocytes, and natural killer cells [165]. IL-6 and IL-8 are also upregulated in many senescent cells and play critical roles in supporting inflammatory environments, such as the primary HGSOC TME. Additionally, HGSOC cells that are cocultured with other cells in the HGSOC TME have been reported to secrete aberrant levels of IL-6 and IL-8. Several of these cells include SKOV3, OVCAR3, OVCAR4, IGROV-1, and HEYA8 [99,166,167,168]. Similarly, senescent IMR90, WI-38, and MSCs and cancer-associated fibroblasts also contribute to amplified cytokine secretion in the primary TME [169,170,171]. IL-6 and IL-8 are highly recognized as critical SASP components and are involved in many pathways that modulate TME stresses and chemoresistance; thus, our review focuses on IL-6 and IL-8.

In vitro studies of IL-6 and IL-8 in HGSOC cells, such as OVCAR3, CaOV3, and SKOV3, have been related to malignant cancer cell behaviors. Induced IL-6 expression via recombination and transfection approaches in non-IL-6-expressing A2780 cells propagated these cells to develop drug-resistant phenotypes [104]. This chemoresistance subsequently enhanced drug resistance genes, such as multidrug resistance 1 (MDR1) and Glutathione S-transferase pi (GSTpi), apoptosis inhibitory proteins, and proteins associated with cell proliferation pathways [104]. Conversely, knockdown of endogenous IL-6 expression in IL-6-overexpressing SKOV3 cells amplified anticancer drug sensitivity [104].

IL-6 can also advance chemoresistance via STAT3 activation. STAT3 is required for HGSOC cell migration and has been established in vitro (i.e., SKOV3 and A2780) to promote expression of proteins that are associated with mesenchymal phenotypes. It also upregulates survival proteins, such as B cell lymphoma-2 (BCL-2) and survivin, that advance chemotherapy resistance [172].

#### 2.6.2. Growth Factors

Many growth factors are overexpressed in HGSOC and play critical roles in ovarian cancer progression and the development of chemoresistant phenotypes. Along with cytokines, many of these growth factors are upregulated in the SASP and exacerbate the alterations in aggressive and chemoresistant HGSOC cell phenotypes. For example, overexpression of the epidermal growth factor receptor (EGFR) and its respective ligands are linked to therapy-resistant HGSOC cell populations [173]. Upregulation of EGFR mediates pathways and activates transcription factors that influence invasive cell behavior, such as survival, proliferation, and migration. Cisplatin-resistant HGSOC cells promote EGFR activity. EGFR signaling through STAT3 led to increased in vitro A2780 cell proliferation, viability, colony formation, and invasive migration [174]. Aberrant EGFR and STAT3 in cisplatin-resistant cells altered cytoskeletal expression and intercellular localization. These cells exhibited increased levels of cortactin and F-actin, where F-actin localized predominately to protrusions [174]. This in vitro work further reported that inhibition of hyperactive EGFR led to cisplatin sensitivity by reducing VEGF and survivin activities. Therefore, genes activated via EGFR signaling (i.e., STAT3) could serve as effective targets for therapeutic strategies in combating chemoresistant cancers [174]. 

Dysregulation of TGF-β1 and TGF-β3 and subsequent downstream canonical and noncanonical pathways have been shown to be activated in refractory ovarian cancer [175]. In fact, TGF-β isoforms are often amplified in the secretome of women who present with therapy-resistant HGSOC cases. Extracellular stress and anticancer therapies modulate TGF-β secretion. It is well-known that TME stress can induce TGF-β-mediated EMT response in HGSOC cells [176]. In vitro studies have also highlighted that inhibiting TGF-β activity via its receptors leads to cisplatin sensitivity in SKOV3 cells [177]. Conversely, cisplatin-resistant HGSOC cells had elevated TGFβR2 levels compared to sensitive cancer cells; this underscores the importance of the TGF-β pathway in chemoresistant HGSOC [177]. TGF-β ligands modulate numerous processes, such as cell growth, differentiation, and SMAD-activated apoptosis. Dysfunctional TGF-β activity in chemoresistant HGSOC cells (A2780) can lead to the activation of downstream genes that are elevated in HGSOCs [178,179].

#### 2.6.3. EVs

Anticancer therapy induces irregular intracellular protein trafficking and EV secretion. In vitro studies have shown that early lysosomes of HGSOC cells pretreated with chemotherapeutic drugs are heavily concentrated with cisplatin [180]. These cancer cells readily export these drugs via endosomal vesicle secretion.

Upon acquiring chemoresistance, HGSOC cells maintain aberrant EV biogenesis and secretion. Transmission electron micrographs showed the cytoplasm of resistant HGSOC cells is more heavily concentrated with EVs than that of chemosensitive HGSOC cells [181]. Chemoresistant HGSOC cell-secreted EVs also encapsulate numerous cytokines and growth factors that were previously highlighted in this review to be important in cell–cell communication and aggressive and chemoresistant cell phenotypes.

Safaei et al. highlighted that in vitro chemoresistant HGSOC cells, OV-2008 cells, exhibited unique EV secretion patterns that sustained their chemoresistant phenotypes by increasing the expression of transporter proteins important in effluxing multiple drugs [180]. These transporter proteins were multidrug resistance-associated protein 2 (MRP2), ATP7A, and ATP7B. They further demonstrated that EVs from chemoresistant cells exhibited a 4.9-fold higher cisplatin content compared to EVs from chemosensitive cells [180], strongly suggesting that EVs play critical roles in maintaining chemoresistant phenotypes.

EVs released by chemoresistant IGROV-1 cells can induce invasion and drug resistance in neighboring bystander cells. HGSOC cells pretreated with EVs from chemoresistant HGSOC cells had increased viability after a subsequent drug treatment, implying that treatment resistance had been transferred with EV treatment [182]. Conversely, cells that were treated with EV uptake inhibitors, such as heparin, dynasore, and amiloride, prior to EV exchange remained chemosensitive. Several proteins that were responsible for chemotherapy adaptation and cell invasion included p38α, p53, and factors involved in JNK signaling processes; p38α and p53 were downregulated, which subsequently contributed to decreased drug sensitivity; c-Jun N-terminal kinase (JNK) pathway activation has been linked to elevated cell invasion and migration. This in vitro study indicated that these EVs could activate prosurvival pathways.

### 2.7. Role of Exosomes in Developing Chemoresistance

Circulating tumor exosomes serve as critical mediators of chemoresistance in the ovarian TME. When exosomes were exchanged in vitro between therapy-resistant OVCAR10 and therapy-sensitive A2780 cells, the sensitive cells developed more invasive and therapy-resistant phenotypes, indicating that phenotypical characteristics could be transferred via exosome content. HGSOC cells pretreated with exosomes harvested from platinum-resistant HGSOC cell lines showed a twofold increase in cell viability after carboplatin treatment. These platinum-resistant ovarian cancer cells further exhibited SMAD4 mutations, which led to some content alterations. In vitro exosome exchange led to increased EMT phenotypes (i.e., increased mesenchymal markers, such as N-cadherin and Zeb1) and induced platinum-resistant phenotypes in once therapy-sensitive cells [183].

Chemoresistant HGSOC cell exosomes can also drive more potent invasive cellular behaviors by promoting angiogenesis. Li et al. showed that exosomes derived from chemoresistant SKOV3 cells increased human umbilical vein endothelial cell (HUVEC) proliferation, migration, and invasion [108]. More strikingly, these chemoresistant cell-derived exosomes led HUVECs to form capillary-like tubes that support angiogenesis. They also reported that chemoresistant cell-released exosomes showed a significantly higher miRNA-130 expression compared to exosomes released by chemosensitive cells. Therefore, exosome miRNA-130 could promote angiogenic development in chemoresistant HGSOC cells [108].

### 2.8. Changes in miRNAs

MicroRNAs are small noncoding RNAs (19–25 nucleotides) that regulate post-transcriptional gene expression through complementary binding and degradation of mRNA and/or translational repression [184]. MicroRNAs are highly conserved across species; however, alterations in their nucleotide sequences or changes in target mRNA can prevent binding to alter gene regulation. Differences in miRNA expression can also alter gene regulation. Numerous in vitro, preclinical, and clinical studies have shown that differential miRNA expression is critical in metastatic initiation, progression, and dissemination [185]. The repertoire of miRNAs in HGSOC samples can be determined from tissue and fluid biopsy samples and can serve as a miRNA signature of various types of ovarian cancers (serous epithelial, mucinous, clear cell). This miRNA repertoire includes critical biomarkers for different cancer types, disease stage, and patient prognosis.

Early clinical studies focused on examining miRNA content in blood and surrounding intraperitoneal fluids for early HGSOC diagnosis. Blood cells and circulating soluble factors that were also present in these samples made it difficult to study the tumor cell miRNA content. Rapid enzymatic miRNA degradation made it further challenging to examine how differentially regulated levels correlated with the disease state. Thus, miRNA content in isolated exosomes has been used to overcome these limitations [186,187,188]. The cholesterol-rich exosome membrane protects sensitive miRNAs from degradation. Circulating exosome miRNA content from the TME more accurately represents molecular phenotypes of tumor-associated cells compared to other signaling molecules.

Numerous factors can alter miRNA content in secreted exosomes, ranging from cellular heterogeneity to differences in the inherent TME stress factors that contribute to exosome biogenesis and secretion. For example, tumor hypoxia enriches miRNA-181d-5p in secreted SKOV3 exosomes [189], and these hypoxia-induced exosomes enhance tumor cell migration and invasion in vitro. Another example of how miRNA content is altered in late-stage tumor-derived exosomes is in the miRNA-200 cohort, a well-acknowledged HGSOC miRNA family, which is elevated in chemoresistant HGSOC patients [190,191]. Conversely, Let-7i/g expression is decreased in chemoresistant HGSOC patients. Reduced levels of Let-7i/g amplified EMT and activated prosurvival pathways in the cells treated with therapy-resistant exosomes. Table 1 further details important miRNAs along with specific gene targets that are significantly regulated that lead to chemorefractory and chemoresistant HGSOC behaviors. Thus, exosome miRNAs are critical HGSOC biomarkers.

## 3. Conclusions

Tumor recurrence is a significant hurdle to overcome across cancers, but is especially poignant in HGSOC. Therapy-induced chemoresistance is well-acknowledged and investigated; yet, the mechanisms describing how properties of the TME lead to cellular stress responses that contribute to chemoresistance in refractory disease remain unclear. One of the main properties of the TME that confers chemoresistance is hypoxia. Hypoxia drives HGSOC populations to develop (p)EMT and CSC profiles. Importantly, the establishment and maintenance of CSC and (p)EMT phenotypes are intimately tied to the induction of autophagy and downstream metabolic reprogramming. Collectively, these events result in metabolically flexible and resilient cancer cells.

It is these populations that have endured environmental stress (hypoxia being only one of many in the TME) that are best equipped to overcome subsequent exposure to stress such as chemotherapy. Zhang et al. identified several gene signatures among treatment-naïve HGSOC populations such as proliferative DNA repair, RNA processing, TCA cycle, among others. Strikingly, however, HGSOC populations post-neoadjuvant chemotherapy were enriched for a stress-associated profile (see Table 2) [56]. Pools of HGSOC cells with a similar stress profile have also been identified as likely sources of relapse-initiating cells [57]. These are two examples of robust studies which were able to differentiate between refractory and resistant HGSOC populations by following the genesis and trajectory of resistance-associated profiles before and after treatment. These studies are necessary given the significant gap in our understanding of the unique characteristics distinguishing between chemorefractory and chemoresistant HGSOC despite the prevalence of both in the clinic. Future studies that are able to identify differentiating properties of chemorefractory and chemoresistant HGSOC can lead to the development of precise treatments for each disease that exploit their unique vulnerabilities.

With regard to chemorefractory HGSOC, key regulators of metabolic reprogramming are potent candidates for future anticancer agents. As we have covered here, metabolic reprogramming is an essential process that governs a cancer cell’s ability to cope with a wide range of environmental stressors, especially hypoxia. Additionally, metabolic reprogramming and the resulting resilient populations are critical contributors to recurrent disease.

There are several approved anticancer agents that target cancer metabolism (e.g., methotrexate, 5-fluorouracil, gemcitabine); however, there are metabolic inhibitors currently in clinical trials that target more newly appreciated key elements of metabolic reprogramming such as glutaminase (CB-839, IPN60090), lactate symporters (AZD-3965), and fatty acid synthesis (TVB-2640) [225]. These metabolic inhibitors are promising therapies, especially for refractory disease. Yet, in order to develop and accurately gauge the efficacy of these therapies against refractory HGSOC, we must drastically improve our diagnostic capabilities—otherwise we will continue to enrich resistant cancer populations with the treatment of standard chemotherapy. 

A candidate feature of HGSOC that would aid in the detection and differentiation of chemorefractory and chemoresistant populations is the secretome. The secretome is greatly influenced by hypoxia since low oxygen concentrations assist in the development of other protumorigenic traits of the TME such as low pH and leaky vasculature. These factors can amplify the secretion of soluble factors and the expression of transcriptional regulators responsible for chemoresistance. Importantly, studies have highlighted that hypoxia modulates pathways necessary for HGSOC cell vesicle synthesis and secretion. Hypoxia-induced exosomes are potent paracrine factors that can induce invasive and chemoresistant phenotypes in neighboring stromal cells and cancer cells. Studies looking at how specific TME stresses contribute to chemoresistant and refractory ovarian cancers through alterations in their secretome profiles are critical in understanding the mechanisms contributing to chemoresistance. Many miRNAs are differentially expressed in chemoresistant phenotypes; however, only a few have been directly linked to chemoresistance [202]. Furthermore, miRNA effects are often studied in isolation, and it remains unclear how the repertoire of miRNAs work together to alter gene regulation in late-stage ovarian cancers. Future studies should investigate these differentially regulated exosome miRNAs individually and collectively to increase our understanding of how they contribute to gene regulation in resistant phenotypes.

Lastly, many chemorefractory components (hypoxia-induced) overlap with the aberrant factors that are present in chemoresistant secretomes (frontline drug-induced) (Table 2). We believe that examining the unique circulating molecules in the chemorefractory secretome is imperative because HGSOC heavily relies on the secretome for metastasis. In fact, these factors support and maintain metastasizing HGSOC cells for subsequent secondary site attachment. Therefore, examining the factors that are present in this type of secretome can lead to new diagnostic measurements that can increase HGSOC detection.

## Figures and Tables

**Figure 1 cancers-14-01418-f001:**
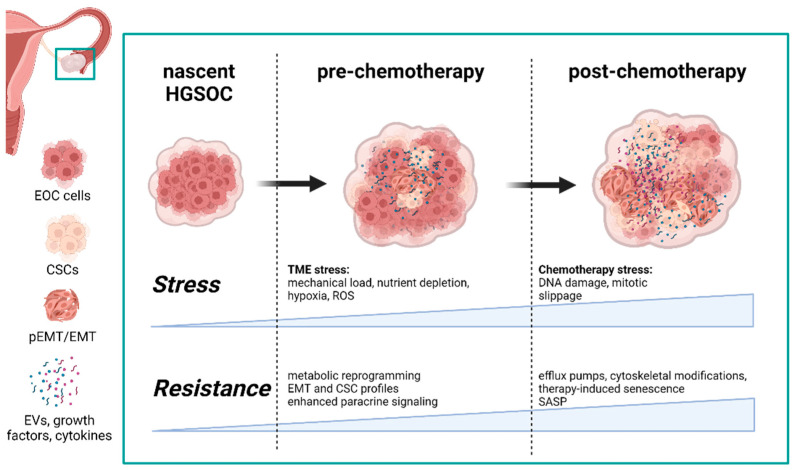
Both hypoxia in the tumor microenvironment and chemotherapy confer resistance to stress and, ultimately, chemotherapies. Each drives distinct changes in the TME composition in terms of cell populations and the secretome. Modifications in the HGSOC secretome as a result of TME stressors not only promote the development of chemotherapy-induced resistance, but are also expanded after exposure to anticancer drugs, thus reinforcing a protective and resilient TME. Abbreviations: (p)EMT—(partial) epithelial-to-mesenchymal transition; CSC—cancer stem cell; HGSOC—high-grade serous ovarian cancer; SASP—senescence-associated secretory phenotype; EV—extracellular vesicles (generated with BioRender, Toronto, Canada).

**Table 1 cancers-14-01418-t001:** Unique and overlapping miRNA profiles in refractory HGSOC. The table highlights the miRNAs and the respective targets/function that were differentially regulated under hypoxic conditions or after chemotherapeutic HGSOC treatment; miRNAs in green represent those that are upregulated and miRNAs in red represent those that are downregulated.

miRNAs	Function and Respective Targets
**Hypoxia**: miRNAs altered in hypoxia/hypoxic tissue and associated functional change	
miRNA-181d-5p [189]	Increased expression in hypoxia-induced EVs; this enhanced M2 macrophage polarization and HGSOC cell migration and invasion
miRNA-940 [192]	Increased expression in hypoxia, HGSOC patient ascites, and exosomes; HGSOC cell–macrophage exosome exchange enhanced M2 phenotype polarization
miRNA-199a-3p [193]	Decreased expression reduced c-Met and AKT activity; this decreased proliferation, adhesion, and invasiveness
miRNA-145 [194]	Suppressed HGSOC; downregulated HIF-1 and VEGF via p70S6K1
**Therapy-induced:** miRNAs altered in chemoresistant HGSOC cells or chemoresistance and associated functional change	
miRNA-93 [195]	Increased expression in chemoresistant HGSOC cells; this altered cell survival mechanisms via PTEN
miRNA-27a [196]	Increased expression in chemoresistant HGSOC cells; this increased MDR and PGP protein expression; inhibiting expression increased cell apoptosis via HIPK2 regulation
miRNAs-130a/374 [197]	Increased expression reduced cisplatin sensitivity; miR-130a knockdown inhibited MDR1 expression and upregulated PTEN expression
miRNA-142-5p [198]	Increased expression enhanced HGSOC cell platinum sensitivity via modulation of antiapoptotic proteins
miRNA-1246 [199]	Increased expression in paclitaxel-resistant HGSOC cells and in patients with severe prognosis; this inhibited CAV-1 expression via the PDGFB receptor and altered cell proliferation
miRNA-221/222 [200]	Increased expression conferred cisplatin resistance via the PTEN/PI3K/AKT signaling pathway
miRNA-433 [201]	Increased expression induced paclitaxel resistance, HGSOC, and poor survival; this modulated HGSOC cell senescence and CDK6 activation
miRNA-891-5p [202]	Increased expression in HGSOC patients and patients who exhibited carboplatin resistance; miRNA associated with DNA repair proteins and MYC regulator genes
miRNAs-200a-c [190]	Increased expression in chemoresistant HGSOC patients; can serve as another diagnostic tool in addition to serum biomarker CA125
miRNA-106a/591 [203]	Increased miRNA-106a expression and decreased miRNA-591 expression in taxol-resistant cells; miRNA-106a targeted BCL-10 and caspase-7; miRNA-591 targeted ZEB1
miRNA-214 [204]	Decreased expression in chemoresistant HGSOC cells; played a crucial role in developing cisplatin resistance via PTEN
miRNA-216b-5p [205]	Decreased expression in taxol-resistant HGSOC cells; overexpression of miRNA and knockdown of SNHG1 led to taxol sensitivity
miRNA-34c [206]	Decreased expression in chemoresistant cells; directly targets SOX9, B-catenin, and c-MYC
miRNA-383-5p [207]	Decreased expression reduced chemosensitivity via TRIM27; this modulated cell proliferation and HGSOC growth
Let-7g [208]	Decreased expression in chemoresistant HGSOC patients; this induced EMT and resistance to platinum therapy
Let-7i [51]	Decreased expression in cisplatin-resistant HGSOC; this activated BRCA1, RAD51, and DNA damage repair pathways
miRNA-29 [209]	Decreased expression in cisplatin-resistant cells; this targeted ECM proteins, such as COL1A1, and modulated ERK1/2 and GSK3B
miRNA-182-5p [210]	Decreased expression in cisplatin-resistant HGSOC cells; this miRNA targeted CDK6
miRNA-134 [211]	Decreased expression in taxol-resistant HGSOC cells; this targeted KAP2 and modulated cell survival and apoptosis
miRNA-6126 [212]	Decreased expression correlated to poor prognosis; highly regulated in exosomes; overexpression reduced angiogenic phenotypes and migration; also acted as a tumor suppressor via integrin β1
miRNA-30a-5p [195]	Decreased expression in cisplatin-resistant HGSOC cells; this elevated apoptosis; exosome miRNA exchange altered chemosensitivity via SOX9
**Overlapping miRNAs**: miRNAs altered in both hypoxic and chemoresistant HGSOC cells	
miRNA-21-3p [213]	Increased expression suppressed HGSOC cell apoptosis via APAF1 binding
miRNA-223 [214]	Increased expression in hypoxia-induced exosomes; this promoted drug resistance in HGSOC cells via the PTEN–PI3K/AKT pathway
miRNA-125b [189]	Increased expression in hypoxia-induced exosomes; this enhanced M2 macrophage polarization and increased HGSOC cell migration and invasion
miRNA-210 [215]	Increased expression enhanced cancer cell viability and proliferation by targeting PTPN1

**Table 2 cancers-14-01418-t002:** Chemorefractory secretome vs. chemoresistant secretome. The table summarizes and illustrates the unique and overlapping factors between the chemorefractory and chemoresistant secretomes; these factors include cytokines, growth factors, genes, and proteins that amplify translational and transcriptional modifications. Many of these components directly contribute to HGSOC progression.

Refractory	Resistant	Both/Not Distinguished
**Genomic**		
	CpG methylation [157,216]	*BRCA1/2* mutation/amplification [59]
		*BRCA1* deletion [217,218]
		*NF1* [218]
		*RB1* [218]
		*CDK12* [218]
		*CSMD3* [218]
		*FAT3* [218]
		*GABRA6* [218]
		*CCNE1* amplification [58,59,218]
		*TP53* mutation [219]
		*IGF2R* deletion [220]
		*MYC* amplification [217,221]
		*MDR1* [59]
		*Rsf-1/HBXAP* [222]
		*NOTCH3* [223]
**Transcriptional**		
*JUN* [56]	*MDR1* [59]	*FOXM1* [218]
*FOS* [56]	*ꞵ-tubulin III* [140]	*NOTCH* [218]
*TNF* [56]	*p38a* [182]	*SNAIL* [183]
*CXCR4* [56]	*GSTpi* [104]	*SLUG* [183]
*SNAI1* [56]	*BCL-2* [172]	*N-CAD* [183]
*VIM* [56]	*Survivin* [172]	*p53* [67]
*GADD45B* [56]	*SMAD4* [183]	*IL6* [56,216]
*MCL1* [56]		
*HIFs* [60]		
*Tetraspanins* [137]		
*SNAREs* [137]		
*Rabs* [137]		
**Translational/post-translational**		
AP-1 [120]	ꞵ-tubulin III [224]	STAT3 [90]
NF-kB [120]	JNK [182]	Akt/mTOR [113]
**Cytokines**		
		IL-6 [110]
		IL-8 [110]
		IL-11 [110]
		IL-17 [110]
		IL-27 [110]
		IL-31 [110]
**Growth factors**		
		TGF-β [104,105,106]
		EGF [104,105,106]
		VEGF [104,105,106]
		TNF-α [118]

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
