# Peer review of "Comparing the Secretomes of Chemorefractory and Chemoresistant Ovarian Cancer Cell Populations"

_cancers, 2022, doi:10.3390/cancers14061418_

Round 1

Reviewer 1 Report

This review by Dawson and colleagues nicely frames the biological changes associated with hypoxia in the biology of ovarian cancer.  The work stresses the adaptations of cancer cells to transient alterations in the tumor microenvironment, and the effect upon secreted proteins and exosomes. In addition to touching on tumor microenvironmental dynamics, there is also a stress on the heterogeneity of individual tumors. The work includes recent progress, which makes it timely. Each of these adds value to the review.  However, there are several weaknesses in the review as well, and a substantial revision is required.

Although the author is a native English speaker, the English is very loose, even misleading, in some cases. For example, the title indicates that the secretome :differentiates between” (suggesting that is the secretome that is performing the action) while the author certainly means that one can differentiate between tumor types based on the secretome.  Certainly the secretome is a product.

Comments such as “recurrence rate is a staggering 70%” need to be supported with a time period.  If this is over the patient’s lifetime, it is not clear how long that might be, and whether it considers the impact of PARP inhibitors and maintenance therapy.  The sentence continues “ is driven by inherent and acquired” and is technically incorrect.  Clinically, a patient that does not respond to chemotherapy due to initial inherent response, is not considered to recur.  Phrases such as “platinum salts crosslink DNA” are strange – the authors likely mean platinum agents induce DNA adducts (only carboplatin and cisplatin are relevant.)  

The manuscript also suffers from conflation at many levels.  The authors treat EOC as if it were one disease, instead of a number of different diseases with very distinct molecular features.  Endometriod, high grade serous, low grade serous, clear cell and mucinous EOC are all very different in their presentation and course.  There is no differentiation among these in the manuscript, and therefore the data is sometimes summarized in a misleading fashion.  The fact that the authors sometimes call out one type of EOC (eg., high grade serous) but then do not in other places lends the reader the impression that these EOC are otherwise the same except where selectively called out, and this is certainly not true.

Similarly, there is a conflation of in vitro, preclinical and clinical studies, so that when a sentence summarizes a number of different parameters induced and it is never clear at what level it has been validated.  This limits the value of the review because these are essential for the reader to understand precisely what phenomena has been characterized in which systems.  Even the cell lines are important and might be mentioned. For example, neither SKOV3 nor HEY8 are particularly appropriate models of high grade serous ovarian cancer (which they are often used for.).  Preclinical models with xenografts will not have a complete cross-talk between tumor cells and immune cells.  This needs to be more clearly delineated across the manuscript.

The authors closely link IL6/8 to SASP, but don’t point out the other roles these cytokines have, particularly in immune evasion (and the notably immunologically cold EOC tumors).   Although there is IL6/8 in the serum of patients, it is not clear precisely which cells secrete it. In general, the cytokine section could be expanded and clarified or summarized in a table if space is limiting. Clarification of the types of cells that secrete the cytokines  (including both tumor and immune cells) would certainly be a welcome addition. 

The clinical significance of hypoxia is well developed, but the presentation is not completely balanced.  On one hand, ascites is not universally hypoxic. On the other, many tumors activate the HIF1a pathway even under normoxic conditions.  The authors might also point out limitations with the approach – at least those so far observed.  For example,  agents that normalize tumor blood flow by pruning tortuous angiogenic vessels (such as bevacizumab) have a modest impact on ovarian cancer patient survival.  They do not reverse chemoresistance.

There is limited in vivo evidence to show that hypoxia is *the* driver in chemoresistance, and entire reviews have been written to try to address the variety of mechanisms that can foster tumor resistance to chemotherapy.

This work follows others recently published in Cancers that focus on hypoxia as a tumor malignancy factor. This particular manuscript is notable for its focus on ovarian cancer, and could represent a useful addition to the field if these issues and omissions are addressed.

Reviewer 2 Report

The manuscript is the review intended to describe the differences in ovarian cancer secretome between inherent and acquired chemoresistance. I would like to point out the following remarks:

1/ do authors mean inherent resistance the same as chemo-refractory cancer? And similarly is according to the authors acquired resistance the same as chemo-resistant cancer?

2/ If the answer is yes, I have a serious doubt that mechanisms of chemo-refractoriness and chemoresistance could be so clearly assign either to hypoxia/EMT/CSCs or to TIS/SASP (as presented on the Fig.1). It is broadly discussed in the literature that CSCs are responsible for chemoresistance and recurrency after front-line chemotherapy. Moreover, hypoxia is not only the feature of refractory tumors. Therefore, authors should present their idea more comprehensively.

3/ The text should be more organized as many information is chaotic, ie. sub-chapter 2 contains many general information about HIF-1alpha, CSCs signaling and EMT process, however, the reader can not find the leading thought connecting these data to the idea of refractory cancer. The same is true for the rest of sub-chapters, ie. paragraph concerning cytokines enumerates several cytokines but only some of them deserved further description. Moreover, there is a convincing argumentation lacking that IL-6 or IL-8 are specifically responsible for chemo-refractoriness.  These remarks relate to the rest of the text. All the manuscript should be rewritten in order to get some clarity.

Round 2

Reviewer 2 Report

There is no doubt that authors have made a progress in the editing of their manuscript. However, in order to get more clarity and factuality I would advise, as follows:

1/ Please add two more figures (one concerning chemo refractoriness and one secondary chemoresistance) that would clearly show what are the differences in secretome in both mechanisms.

2/ This differences could also be summarized in table showing what functions for chemo refractoriness and chemoresistance have particular cytokines, growth factors, hypoxia, acidosis etc. All differences between chemo refractoriness and chemoresistance should be asserted. As IL-6TNF-alpha and IL-8 are so important in both chemo refractoriness and resistance, are there any differences in their function between both mechanisms?

3/ Please use the terms “refractoriness” and “resistance” properly in the text as they are in some places used interchangeably (ie. line 474 in the chapter about resistance authors write about the role of EGFR in refractoriness).

4/ Chemo refractoriness could result from specific HGSOC cancer genome. The genetic characteristics of HGSOC having influence on chemo refractoriness should be discussed in the beginning of chapter 2.

5/ Some sentences are too long and complex, therefore need change of style. Some are also not fully understandable (ie. lines 91-94, 115-119, 346-352, 355-361, 484-488 and some more)

6/ Please define shortly the mechanism of autophagy and mitophagy. Similarly, please define what is cellular dormancy and senescence (including OIS and TIS) (in the places in the text they are firstly mentioned).   

7/ When you write about IL-11, -27,- 31 cytokines in line 259, please devote them at least one sentence/ each in the following text. Moreover, as ovarian cancer is an inflammatory tumor, some information about IL-17 should be added.

8/ Conclusion should contain the authors’ personal opinion about the differences in both refractoriness and chemoresistance and their meaning for modification of the approach to the therapy of ovarian cancer

Round 3

Reviewer 2 Report

Two points of minor concern:

1/ Please add some sub-headings to the longer paragraphs in order to support clarity and fluency of reading.

2/ I asked before to write some propositions of how to act against chemorefractoriness (mainly drived by hypoxia) therapeutically. Could you make such predictions in the Conclusion chapter? The sentence 583-585 is not proper as clinical trials with anti-EGFR moAbs did not show satisfactory results.  
